



# Detecting the Melting Layer with a Micro Rain Radar Using a Neural Network Approach

Maren Brast[1] and Piet Markmann[1]

[1]METEK Meteorologische Messtechnik GmbH, Fritz-Straßmann-Str. 4, 25337 Elmshorn, Germany

**Correspondence:** Piet Markmann (markmann@metek.de)

**Abstract.** A new method using the Micro Rain Radar (MRR) to determine the melting layer height is presented. The MRR is a small vertically pointing frequency modulated continuous wave radar which measures Doppler spectra of precipitation. From these Doppler spectra, various variables such as Doppler velocity or spectral width can be derived. The melting layer is visible through a higher reflectivity and an acceleration of the falling particles, among others. These characteristics are fed to a neural network to determine the melting layer height. For the training of the neural network, the melting layer height is determined manually. The neural network is trained and tested using data from two sites covering all seasons. For most cases, it is well able to detect the correct melting layer height. Sensitivity studies show that the neural network is able to handle different settings of the MRR. Comparisons to radiosonde data and cloud radar data show a good agreement in melting layer heights.

## 1 Introduction

The bright band in radar meteorology shows the location of the layer where solid precipitation melts into rain in a complex process. Near the freezing level, aggregation leads to larger ice particles. When melting begins, the particles are coated by liquid water, appearing as very large raindrops, until all ice is melted and the particles collapse into raindrops. The reflectivity maximum is partly due to the different dielectric factor of ice and water. At the bottom of the melting layer (ML), the particle density decreases due to acceleration and the smaller size of the particles, resulting in a decreasing reflectivity (e. g., Austin and Bemis, 1950; Battan, 1959).

Though the bright band has been detected since the beginning of radar meteorology in the 1940s (e. g., Byers and Coons, 1947), it has not yet been fully understood. The detection of this layer is important for various applications, e.g., for the correction of precipitation estimation or the prediction of the kind of precipitation at the surface. Knowledge of the ML is also important for aviation. Especially near airports it is very useful to know if there might occur icing. A typical rain event with rain at the surface and snow above is not a concern for airplanes. If, however, it is raining and the temperature is below $0\,°C$, the supercooled falling rain can turn to ice once it hits a surface such as an airplane. The accumulating ice causes the shape





of the plane to change, disturbing the aerodynamic properties of the wing. The detection of the ML at airports can thus help assess the risk of icing.

Many previous studies have used various radars to detect the ML. Some used radars scanning a volume (e. g., Gourley and Calvert, 2003) or radars pointing vertically (e. g., Fabry and Zawadzki, 1995; White et al., 2002; Johnston et al., 2017) with
different wavelengths, using reflectivity data or a combination of reflectivity and velocity data, others use polarimetric radars, using the properties of the echoes to distinguish between snow and rain (e. g., Giangrande et al., 2008). In contrast, this study uses a smaller and less expensive remote sensing instrument, the Micro Rain Radar (MRR) by Metek GmbH, which has been widely used to measure vertical profiles of precipitation by, e. g., Löffler-Mang et al. (1999); Peters et al. (2005); Yuter et al. (2008); Kneifel et al. (2011); Maahn and Kollias (2012). It is a frequency modulated continuous wave radar pointing vertically,
operated at 24 GHz. Due to its compact size, it can be easily installed virtually independent of site conditions and measures Doppler spectra with high accuracy (METEK GmbH, 2018). From the spectra, it calculates two classes of variables. While the first class consisting of the attenuated, equivalent reflectivity factor (ZEA), the Doppler velocity (VEL), and the spectral width (WIDTH) is just a condensed description of the spectral properties, the second class consisting of drop size distribution, path integrated attenuation, Rayleigh-radar reflectivity factor, liquid water content, and rain rate (RR) represents a retrieval of
physical target properties under the assumption that the backscattered signal is caused solely by rain drops. Although variables of the last class have no physical meaning in case of frozen or mixed precipitation, they may act as indicator for the presence and height of an ML.

The MRR has been used previously to detect the ML height. Perry et al. (2017) use the gradients of ZEA and VEL to detect the ML top and bottom, respectively. Cha et al. (2009) detect an ML where the largest positive and negative vertical gradient of
the rain rate embrace the maximum rain rate. A large dataset was used, excluding the winter months with surface temperatures below 0 °C. Pfaff et al. (2014) have compared this algorithm to two other algorithms: fitting an analytical function to the reflectivity profile, and combining reflectivity and falling velocity to derive the ML height. They conclude that the combination of reflectivity and falling velocity gives the best results. However, their data consists of two case studies. For operational use, a much larger dataset should be used for testing.

This paper presents a new method to extract the ML height from the MRR data which can be used operationally in all weather situations. It uses a neural network approach, which is well suited to nonlinear and complex problems and which does not rely on certain assumptions of the data distribution. A neural network learns from examples and has no need of imposing fixed thresholds or the shape of reflectivity and/or velocity profiles such as previous studies (e. g., Fabry and Zawadzki, 1995; White et al., 2002; Giangrande et al., 2008; Cha et al., 2009; Perry et al., 2017). Given the right training, this approach is
therefore more flexible and able to generalize and thus detect unusual melting layers such as melting layers on the ground or two melting layers at the same time. It is also possible to design a neural network in different ways, so that it is for example able to either detect a specific height such as the height where melting starts or to detect the whole vertical extent in which the melting occurs. Our neural network (NN) was designed to detect the whole melting layer and was trained and tested using data with a high temporal resolution and including all seasons and different precipitation types such as rain, snow, and sleet.





Sect. 2 presents the neural network approach. It describes how the training data is generated and the setup of the NN. In sect. 3, the performance of the NN is described. It demonstrates how the performance of the NN is assessed, which kind of situations it handles well, and what are the limits of this approach. The last section, sect. 4, gives a discussion and a conclusion.

## 2 Method

5 The neural network approach is well suited for complex nonlinear problems, which meteorological phenomena often are. Though the NN lacks all physical basis, it can yield accurate results (e. g., Marzban and Stumpf, 1996; Liu et al., 2001). A difficulty with NNs is the need for training data. During training, an NN is given the input data as well as the desired output. Through this process it learns patterns and can subsequently be applied to previously unknown data. In the following, the measurement data used is described. Then, the method of generating the training data is presented, followed by the description 10 of the NN, the training process and the post-processing.

### 2.1 Measurement data used

For this study, most data was measured by two MRR-PRO by Metek GmbH (which is a further development of the MRR-2), deployed by the German Weather Service in Germany at the airport in Hamburg in the North German Plain and at Hohenpeißenberg, a topographically isolated mountain of almost 1 km height. The measurements from Hamburg were taken from 15 November 2017 to April 2018 with an interruption in the first half of January due to technical issues. The measurements at Hohenpeißenberg range from August 2017 to December 2018 with interruptions from December 2017 to April 2018 and in May 2018. The measurements were taken at 128 range gates with a vertical resolution of 15 m in Hamburg and 25 m at Hohenpeißenberg and with 10 s time resolution at both sites. Only days with precipitation were used, resulting in 166 days from Hohenpeißenberg and 90 days from Hamburg. Within the course of a day, precipitation can change substantially. Therefore, 20 the days were subdivided into four intervals of 6 hours. This makes the selection of training data more flexible and avoids including long time spans of no precipitation when a short period of rain is desired to be used, and at the same time avoids the data being split into too many very small intervals which are impractical to process.

Additional data was measured in Elmshorn by Metek, to test the sensitivity of the NN to other height resolutions. One day with a resolution of 50 m and four days with a resolution of 100 m are available.

25 The dataset covers many different situations, including cases without precipitation, cases of light drizzle and snow without MLs, high and low MLs, MLs on the ground with sleet at the surface, and even rare situations with hail or two MLs at the same time.

### 2.2 Generating training data

Training an NN requires training data with knowledge of the desired output. This means that reference data of the true location 30 of the ML is needed. One option of determining the ML is to look at the height of $0\,°C$ in a temperature profile. Temperature could be measured by a mast, which can only provide data up to limited heights, or by radiosondes, which are restricted to





few sample points on the time axis. Airplanes flying through the ML could provide very detailed data but would be completely impractical for generating a large set of training data. Data from models are available at a high spatial and temporal resolution, but due to inherent uncertainties they cannot be relied upon to provide a correct height of the ML at the exact location of the radar with a resolution in the order of seconds, because the ML height can be very variable in time and space. Therefore, a
different method of finding the true ML height must be used.

Looking at time-height-series of ZEA, VEL and other output from the MRR, the human eye is well able to detect the ML height in many situations (e. g., by the increase in VEL or the maximum in RR or ZEA). Therefore, a tool was build to draw the upper and lower boundaries of the ML into plots from different output variables of the MRR by moving the mouse over the plot by hand (see Fig. 1). This has the advantage that the "true" ML is determined with the appropriate time resolution
at the location of the MRR. Different variables show different properties of the ML. Some variables show a stronger gradient at the upper boundary, while the gradients of other variables are stronger at the lower boundary of the ML. For example, the reflectivity shows the strongest gradient at the upper boundary of the ML because it has a maximum where the particles are already coated with water but are still large because the ice has not melted completely yet. The velocity grows toward the lower boundary of the ML when the air resistance decreases and the density of the particle increases. Therefore, the upper and lower
boundary of the ML were determined by different variables.

Figure 1 shows the five variables which are used to draw the ML height. For each variable, two lines were drawn (upper and lower boundary) in the same color. For the upper boundary, the RR and the gradient and curvature ZEA' and ZEA" (first and second derivative of ZEA) were used, while the lower boundary was determined by WIDTH and the gradients VEL' and ZEA'. The plot shows that the drawn lines can deviate considerably for the five variables, which is why not all variables are
used to determine both the upper and lower boundaries of the ML. The lines from the variables chosen for physical reasons lie fairly close together, giving confidence to the choice of variables. This choice results in the optimal case in two triplets of lines for the upper and lower boundary of the ML. The drawing of the lines has an inherent uncertainty, stemming on the one hand from the ability of the human eye to detect the pattern of the ML and on the other hand from the imperfect movements of the mouse. Therefore, the standard deviation of both triplets was calculated. The transition in the upper and lower boundary
of the ML, describing the uncertainty, are determined by +- the standard deviation centered around the triplet averages. Within the transition the uncertainty of the procedure is expressed as a linear function between 0 (no ML) and 1 (certain ML). If there are less than three lines available, the uncertainty is not determined by the standard deviation but by fixed values which grow with decreasing number of available lines. The last panel in Fig. 1 shows the resulting "true" ML in gray, overlain with the lines drawn from the five variables. The dark gray area depicts the region where there is certainly an ML and the light gray
area depicts the transition region. The training data for the NN thus consists of one profile for each measuring time, with values ranging between 0 and 1.

The ML can only be determined by eye for cases where the ML is fairly continuous in time, therefore only such events were chosen to generate the "true" data. Many cases consist of fairly continuous rain at the surface and a well developed ML above, but the training data also includes MLs at the surface, cases with no ML due to snow or drizzle, and cases without
any precipitation. Also, some cases with showers were included, where no ML was detected by eye and the desired output is



"no ML". In total, 79 intervals of 6 hours were drawn, giving a total time of 474 hours. An overview of the cases is given in Table 1. Within one interval, there can be more than one situation. In many cases, precipitation starts or stops, so that the cases were classified on the basis of the most significant characteristic (e.g., "ML above ground" even if half the time there is no precipitation).

**Figure 1.** Determining the "true" ML for Hamburg on 8 Mar 2018, 15:00 to 18:00 UTC. Shown are WIDTH, RR, ZEA', VEL', ZEA" and the "true" ML. The manually determined ML heights from each variable are shown as colored lines: blue lines from WIDTH, orange lines from RR, purple lines from ZEA', red lines from VEL', and green lines from ZEA". The resulting "true" ML is shown at the bottom right as shaded gray (see text).





**Table 1.** Overview over Training Cases of 6 hour intervals

| no precip | snow or drizzle | ML above ground | ML on ground | showers |
|-----------|-----------------|-----------------|--------------|---------|
| 3 | 14 | 48 | 7 | 7 |

## 2.3 Neural network

NNs are modeled after the human brain. The idea is to connect individual artificial neurons and train them for a specific task. A neuron receives input data, combines the weighted input and gives an output according to an activation function. The weights determine the importance of the input data and the activation function determines the level of activity of the neuron. The output value is only transmitted if a threshold is exceeded. In a simple NN such as used here, the neurons are arranged in an input and an output layer, with optional hidden layers in between, and the data only moves forward through the NN. Each neuron is connected to all neurons in the previous and the next layer. The connections between the neurons have a weight, which is random in the beginning and adjusted during the training process to give the desired output. The training process is performed by giving data to the NN multiple times. Each time, the output of the NN is compared to the desired output, an error is calculated and the weights are slightly adjusted to minimize the error. Once the NN is trained, it is evaluated with previously unseen data.

As input data, profiles of both ZEA and VEL proved to be the best choice. The combination of these two variables have also produced the best results in Pfaff et al. (2014). Before using these profiles within the NN, they need to be prepared. First, the data was interpolated to a vertical resolution of 25 m, and aliasing of VEL was corrected. Also, where ZEA falls below a threshold of -5 dBZ, ZEA and VEL were considered to be invalid. Because the NN is not able to process missing values, they were set to 0 for VEL and to -10 for ZEA. Since the NN needs the values of the input to roughly range between 0 and 1, ZEA and VEL were scaled accordingly. To use the fact that the height of the ML shows some persistence in time, profiles of ZEA and VEL from four previous time steps were also used as input. Those four profiles were not taken from the time steps directly before, but were spaced apart 6 time steps to be able to "see" farther into the past without having to use all 24 profiles. Then, to account for the fact that most MLs in the training dataset are located roughly in the middle of the profiles, and to increase the volume of training data, the profiles were subdivided. From each profile, the bottom 64 heights were taken as one sub-profile. Then, the interval of 64 heights was shifted by one height, and these 64 heights were taken as another sub-profile. This procedure was continued to the top 64 heights, resulting in 65 sub-profiles. This improved the prediction of MLs at the very bottom and top of the profiles. This whole preprocessing of the data results in an input array consisting of sub-profiles of multiple time steps of both ZEA and VEL, yielding 2*65*64*4=33280 values for each measurement time step of the MRR.

The NN used here is a feedforward NN with two hidden layers. Each hidden layer has 64 neurons, the learning rate at the beginning is 0.002. The python library scikit-learn (Pedregosa et al., 2011) was used to develop the NN. Due to the large amount of training data, it was impractical to load all data into memory at once, so it was determined manually in which order and how often the individual datasets were used in training. To be able to determine the performance of the NN during training, validation data was used. The training procedure of the NN began with iterating through all 6 hour intervals of the training





dataset once, successively in a random order. After each 6 hour interval, the performance of the NN was determined with the validation data and the mean square error (MSE) between the prediction and the "true" ML was calculated. For the calculation of the MSE the prediction was slightly tweaked for false positive values to nudge the NN toward a conservative prediction. After one epoch, the mean MSE was calculated, and all 6 hour intervals with an MSE larger than the mean MSE were not

used for training in the following epochs. This means that for each epoch, the amount of training data decreases, until there is no more data left, denoting the end of the training. After each epoch, the learning rate was divided by two and the remaining data was again shuffled. If the mean MSE of one epoch was worse than that of the epoch before, that epoch was not used for training.

     The validation data consisted of seven 6 hour intervals, chosen to represent different situations such as an ML above the

ground, an ML on the ground, a case with showers, and a case of snow. The common method of splitting all data randomly into training and validation data seemed impractical here, because it was important to cover many different meteorological situations within the validation set to ensure that the NN can handle them all well. Picking the validation data randomly would either result in not covering different situations, or the random picking would have to be applied to individual profiles instead of 6 hour intervals. This, however, was impractical due to memory constrictions and turned out not to be necessary.

To ensure that the NN is not overfitted to the validation data, the rest of the measurement data, for which no "truth" was created, was used as test data. Since the NN should handle all possible situations, the performance of the NN was assessed visually for the test data. The NN was adjusted and the training process was repeated until the visual inspection showed a satisfactory result (see also sec. 3).

     After the training was complete, there occurred situations where an ML was wrongly detected. Many of these situations had

similar characteristics. These cases were situations with a high ML and a wrongly detected second ML near the surface. Also, sometimes the upper edge of valid MRR data was wrongly taken as ML. Therefore, a post processing step was included to remove the wrong detections in these specific cases, based on fixed thresholds of rain rate and signal to noise ratio. Also, MLs that lasted less than six time steps were removed to avoid short term clutter, and MLs with a confidence of less than 0.2 were set to 0.

## 25    3    Results

This section describes the performance of the fully trained NN, starting with a straightforward case of a well developed continuous ML. Then, more complex cases are shown to give an overview over the range of possible situations and to show how the NN is able to handle these situations. Also, sensitivity studies of vertical and temporal resolutions and comparisons with an ML detected by weather radars, with radiosonde data, and with a cloud radar are shown.

### 30    3.1    Performance of NN

An example of the performance of the NN is given in Fig. 2 for a well developed ML. In the time period shown, there is a continuous ML at about 1900 m and rain below. The output of the NN is shown in the range between 0 and 1, indicating the





uncertainty with which the NN detects the ML. Values close to 1 indicate a high confidence in the existence of an ML. The ML detected by the NN is fairly broad. This is due to the training data, where the ML starts at the top with completely frozen particles and ends only when all particles are liquid. This means that the NN will also give broad MLs. The NN could as well have been trained with one height of the ML, e.g., the middle of the ML height; it is a choice of design. The thickness of the

5  ML is also influenced by the precipitation intensity, in the training data as well as detected by the NN in the test data. This behavior has previously been observed by, e. g., Klaassen (1988).

Though the performance of the NN in the situation shown in Fig. 2 is obviously good, the NN needs to be validated for many different situations if it is to be used operationally. However, calculating a metric to quantify the error is difficult, since common metrics such as mean square error or probability of detection all need to know the expected outcome. Sect. 2.2 described the

10  procedure with which "true" data was generated, which was needed for the training of the NN. We wanted to use as much data as possible for the training process because the performance of the NN depends on the amount of training data. Therefore, most of the data where a "truth" was generated was used to train the NN and the rest of the "true" data was used for validation. During training of the NN, it was constantly tested against a validation data set and the mean square error was calculated (see sect. 2.3), which determined when the training is complete. For testing the NN, no "true" data was used, since creation of "true"

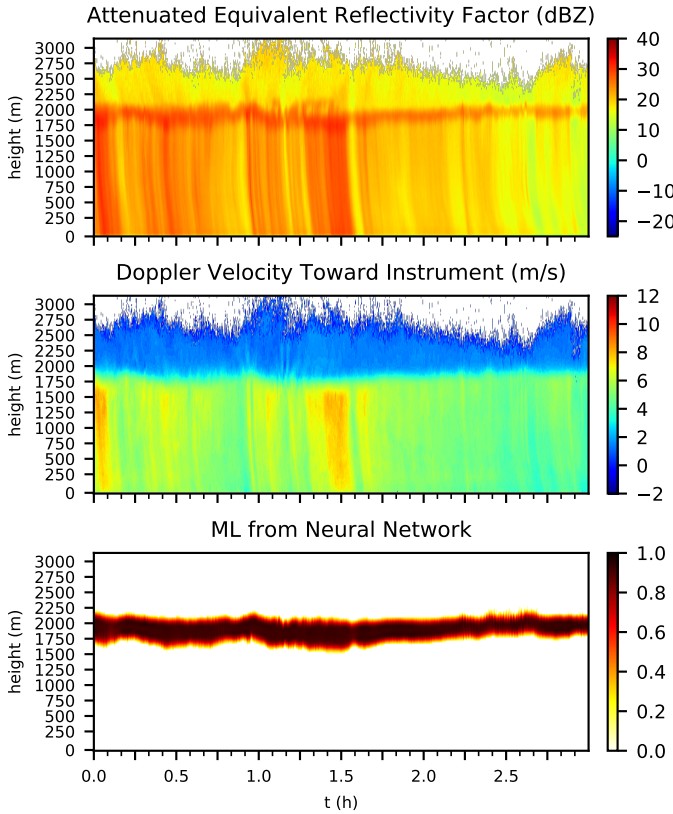

**Figure 2.** ZEA, VEL, and ML detected by the NN for Hohenpeißenberg on 13 Jun 2018, 0:00 to 3:00 UTC.





data is time consuming and only possible for situations where the eye can easily identify an ML. Also, we wanted the NN to be conservative, meaning that a false negative is considered less severe than a false positive. Therefore, during the development of the NN, the performance of the NN was visually inspected and a low MSE was not automatically considered a good result. The final evaluation of the NN was conducted by plotting the output of the NN as well as the time series of the profiles of ZEA,

5    VEL, and others. Then, all available 261 days of data were visually inspected and evaluated. In the following, a few cases are shown which are exemplary of the performance of the NN in different weather situations.

Figure 3 shows a case without ML. The temperature on the ground is below zero and the precipitation is snow throughout the measuring range. ZEA has no maximum and VEL shows no acceleration. The NN correctly detects the absence of an ML.

Figure 4 shows a case where the ML lifts from the ground. Surface observations measured a temperature at 2 m height of

10   around 1.5 °C at 0 UTC, increasing to about 3.5 °C at 3 UTC, and the precipitation at the surface turned from sleet to rain. In the beginning, the snow has started to melt, but the melting process has not completed above ground. With increasing temperature, the melting starts higher up and is complete before the precipitation hits the ground.

Figure 5 shows a convective case with showers. In these situations, the ML is not continuous in space and time as it is in a case with a homogeneous cloud layer and stratiform rain. Instead, in convective cases vertical motions can be considerable

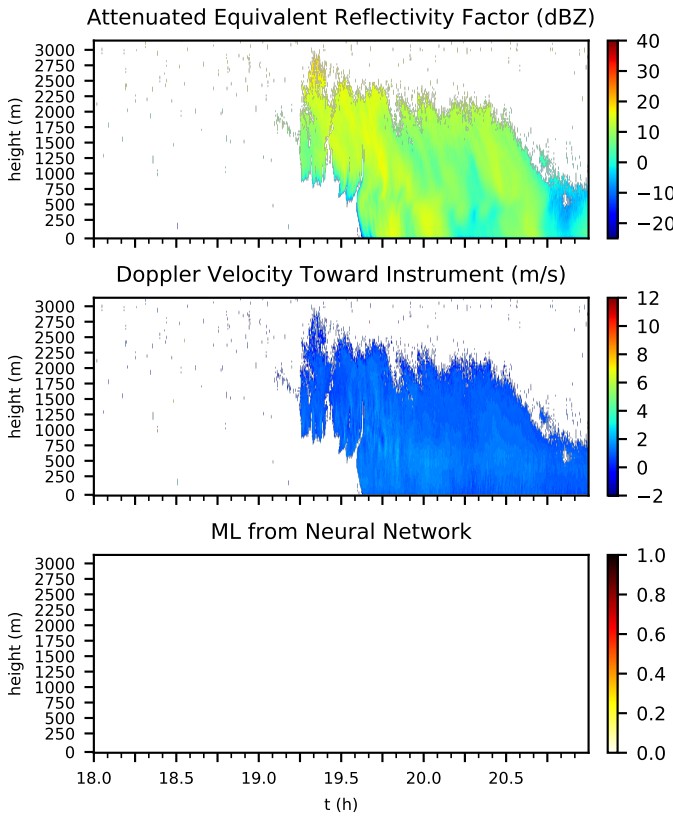

**Figure 3.** Same as Fig. 2 for Hohenpeißenberg on 18 Nov 2018, 18:00 to 21:00 UTC.





and can distort the ML. Vertical motion can lift raindrops inside a cloud to heights where the temperature is well below zero, and downdrafts can accelerate frozen particles toward the ground. Therefore, in such cases the detection of an ML by eye is difficult or even impossible. Accordingly, the NN has difficulties dealing with these cases. During the training process, the NN was tuned to detect as little as possible in these cases, but it was not possible to train the NN not to detect anything

5   without impeding the abilities of the NN for other cases. Therefore, convective cases are those cases where the NN has the most problems.

In our dataset we found two cases where two MLs at the same time were clearly visible, one of which is shown in Fig. 6. At Greifswald, the radiosonde at 12 UTC measured two layers with temperatures above freezing, one freezing layer in between and freezing temperatures above. The two MLs are visible to the eye, especially in the reflectivity data, and also observed by

10  the NN, despite the fact that there was no such case in the training data set. This is a sign that the NN is able to generalize.

## 3.2   Sensitivity studies

For the NN to work operationally, it must be ensured that it can handle various settings of MRR operation parameters. Therefore, four 6 hour intervals with precipitation from an MRR located in Elmshorn with a vertical resolution of 50 m and ten 6

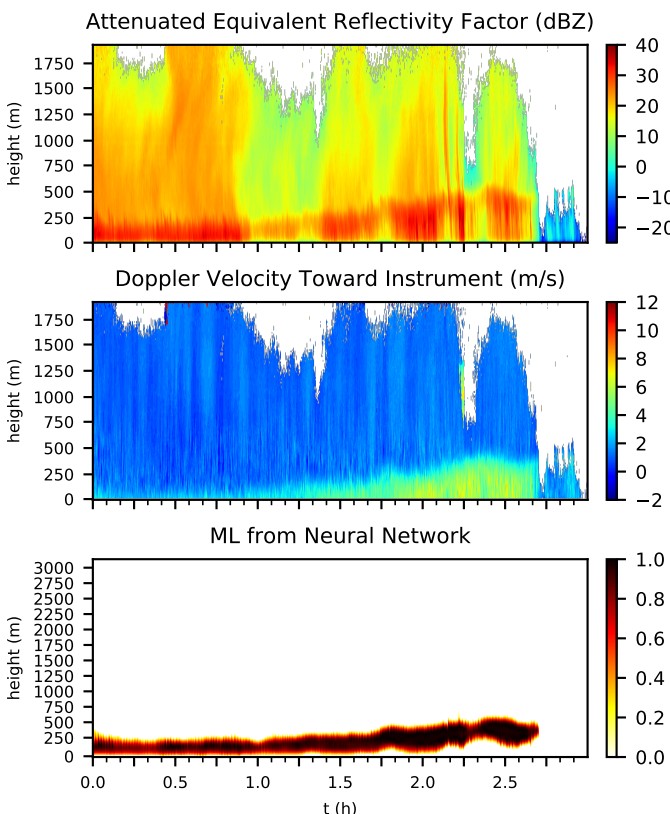

**Figure 4.** Same as Fig. 2 for Hamburg on 14 Dec 2017, 0:00 to 3:00 UTC.



hour intervals with a resolution of 100 m were tested. Since all data is first interpolated to a resolution of 25 m and the NN expects 128 height levels, no ML can be detected above 3200 m. Since the ML heights in the data from Elmshorn are below this height, the NN was well able to detect the ML, proving that it can handle data with various vertical resolutions up to 100 m. For even coarser vertical resolutions, the NN might have trouble detecting an ML since then the resolution might be of the

5 same order or even larger than the vertical extent of the ML.

Also, the sensitivity of the NN to the temporal resolution of the data was tested. The NN was trained with a temporal resolution of 10 s and with profiles of five different times each one minute apart from the next profile. Since the NN should work for different operational settings of the MRR, it should also give good results when an MRR is operated with a larger averaging interval. Therefore, part of the test dataset (20 days) was averaged over 30 s and 1 min, and the NN was again

10 given five profiles each one minute apart from the next. The output of the NN was then visually inspected. Only averages up to one minute are tested because MRRs are seldom operated with longer averaging intervals. The difference between the different averaging intervals are small (not shown). In general, for a longer temporal average, the detected ML gets smoother. In some cases, the NN with a larger averaging interval wrongly detects an ML with a low probability for short periods of time ("clutter"). Also, the edges of the ML are a bit less clearly defined, manifested by slightly larger tails of a low probability of

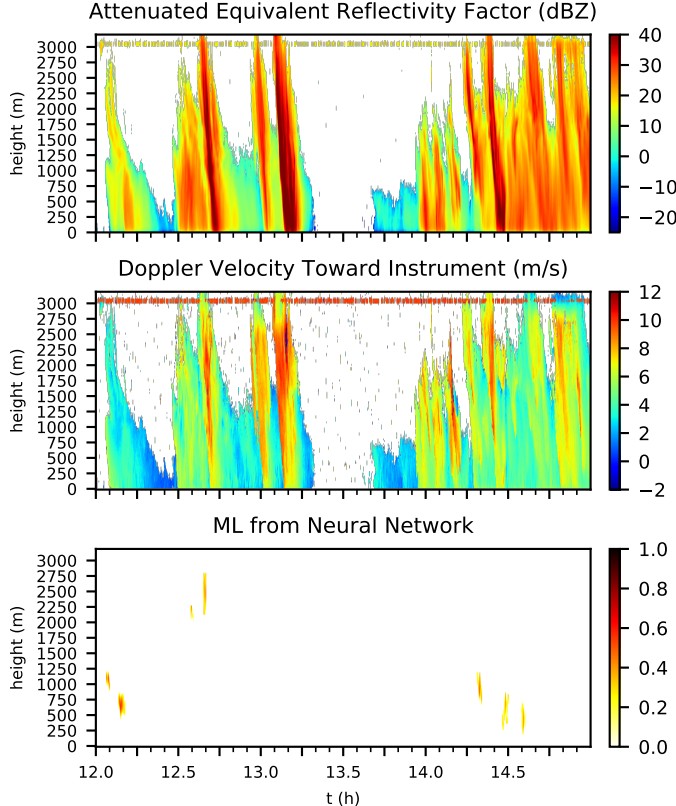

**Figure 5.** Same as Fig. 2 for Hohenpeißenberg on 31 Aug 2017, 12:00 to 15:00 UTC.





detected ML. However, in some cases there are small gaps in the detected ML for the short averaging interval of 10 s when the ML is detected in some time steps but not in others. This results in small fragments of correctly detected ML which might in addition be filtered out by the criterion that MLs lasting less than six time steps are removed. For larger averaging intervals, these gaps are sometimes filled because of the smoothing of the data. The described differences only appear for cases where

5   the detection of an ML is somewhat difficult due to, e. g., showers, whereas for most cases, differences between the different temporal averages are hardly discernible. Therefore, we conclude that the NN is able to handle different temporal resolutions of MRR data up to a resolution of one minute.

### 3.3   Comparison to C-band radar

The ML height detected by the NN was compared to the ML height detected by C-band radars deployed by the German

10   Weather Service (DWD). The DWD is in the process of developing an ML detection algorithm for the operational weather radars in Germany, supported by the numerical model COSMO-D2. For the two locations of the MRR where most of the used data comes from, the ML was compared to the ML detected by the surrounding C-band radars. For the MRR at Hamburg airport, the surrounding weather radars are located at Boostedt, Rostock, and Hanover. For Hohenpeißenberg, they are located

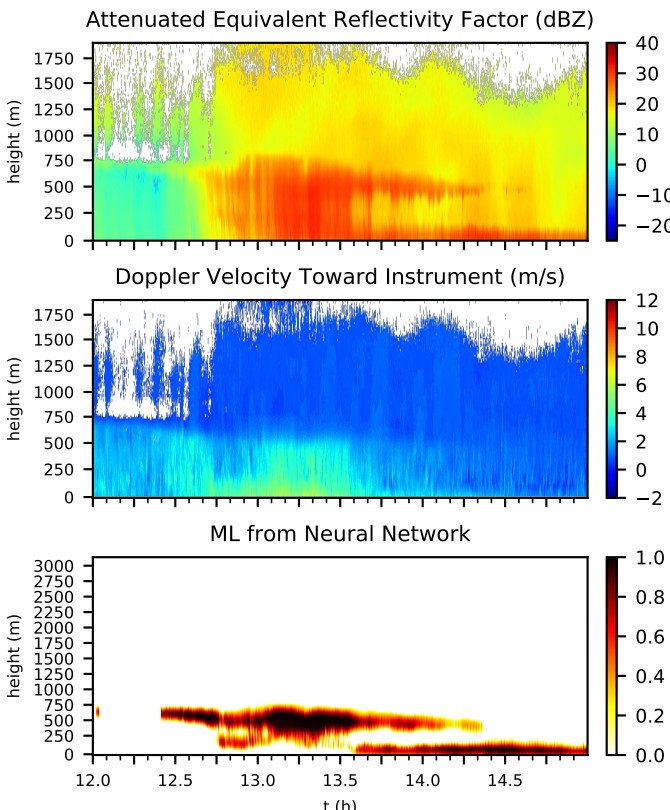

**Figure 6.** Same as Fig. 2 for Hamburg on 31 Mar 2018, 12:00 to 15:00 UTC.



at Türkheim, Isen, and Memmingen. Data is available until June 2018. For 35 (18) of the 90 (84) days where the MRR detected precipitation in Hamburg (Hohenpeißenberg), at least one weather radar in the vicinity of the MRR detected an ML at the location of an MRR. Many cases were detected by the MRR but not by the weather radars. Mainly, this is because the weather radars are too far away from the MRR.

5    Figure 7 shows an example of the MRR data with the ML detected by the weather radars overlain as dots on the output of the NN. The C-band radars do a volumetric scan every 5 minutes, therefore the temporal resolution of the detected ML is much coarser than that of the MRR. In the shown case, the different weather radars disagree on the exact height of the ML, possibly because the weather radars measure with a low elevation angle and are located between 40 km and 90 km away from the MRR, so the spatial resolution is not as high. Also, the coarser spatial resolution could explain the differences to the MRR, since the

10    scanned volume might be different. The MRR seems to be much better suited to detect the ML height at one location, thus the information of the MRR might aid the detection algorithm of the weather radars.

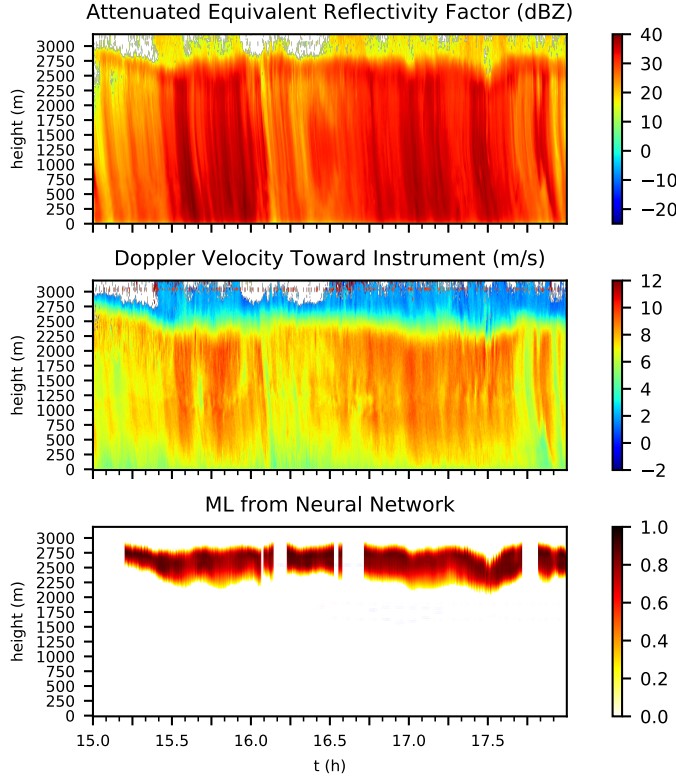

**Figure 7.** ZEA, VEL, and ML detected by the NN for Hohenpeißenberg on 10 Aug 2017, 15:00 to 18:00 UTC. The dashed lines indicate the top and bottom of the ML heights detected by three weather radars: Green from Isen, blue from Memmingen, pink from Türkheim.



### 3.4 Comparison of ML height to radiosondes

The ability of the NN to determine the ML height has mostly been assessed by visual comparison with the other data measured by the MRR (see sec. 3.1). As explained in sect. 2.2, it is difficult to find other sources of the ML height with which a meaningful comparison can be made. At Hohenpeißenberg, radiosondes are launched about twice a week to measure ozone. These radiosondes also measure temperature, so that a comparison can be made between the height of the freezing level of the radiosonde and the top height of the ML detected by the NN, since the frozen particles should start to melt at the level of $0\,°C$. Such a comparison has limitations, because for those times when a radiosonde is launched, precipitation must be detected by the MRR within a short time period of the launch. Also, the weather situation must be fairly constant so that a sudden change in ML height is unlikely and it can be assumed that the ML height detected by the MRR can be compared to the $0\,°C$ height of the radiosonde. When sorting through the available radiosonde data, cases were rejected where it was clear from MRR data that the precipitation and therefore the ML height might be changing too quickly or the duration of the precipitation was less than two minutes. From all available data from Hohenpeißenberg, 18 cases were found for which a fairly reasonable comparison can be made. Figure 8 shows a comparison between the freezing level of the radiosonde and the top height of the ML determined by the NN. Since the time difference between the detected ML by the NN and the measurement of the radiosonde is up to about 5 hours and the radiosonde drifts horizontally as it rises, the value of this comparison is limited. Nevertheless, overall there is a good agreement between the radiosondes and the NN, almost independent of the time difference between the ML measured by the NN and the radiosonde. The largest difference between ML top and freezing level is about $320\,m$ for a case where the time difference between both measurements is about 3,5 hours. In four cases, the radiosonde measured negative temperatures at the surface and the NN correctly detected no ML (blue circles at height $0\,m$ in Fig. 8).

### 3.5 Comparison of ML height to cloud radar

For four additional days in December 2018 and January 2019, a comparison between the ML detected by the NN and a cloud radar (MIRA-35 by Metek) was made. The cloud radar determines a melting layer height to identify plankton. The ML is determined from the linear depolarization ratio (LDR) in the vicinity of the ML taken from a database (METAR-data, radio soundings, or model data). Figure 9 shows an example of this comparison. The background shows ZEA measured by the MRR for reference. The gray area denotes the ML detected by the NN, and the blue line is the ML detected by the cloud radar. The ML from the NN is fairly broad and encompasses the maximum of the reflectivity, as described in the previous sections. The ML from the cloud radar mostly follows the bottom boundary of the ML detected by the NN. This is caused by the way the ML is determined by the cloud radar. The LDR is sensitive where the particles are already mostly melted, while the NN considers the whole process of melting. Considering these differences in purpose and determination of the ML, the agreement between both instruments is very good.





## 4    Conclusions

We have developed a neural network to detect the ML from MRR data operationally. The available data encompasses measure-
ments from all seasons and from two MRRs at two locations in Germany, one location being on an isolated mountain in the
very south of Germany and the other in the North German Plain. With this dataset, the NN was tested extensively.

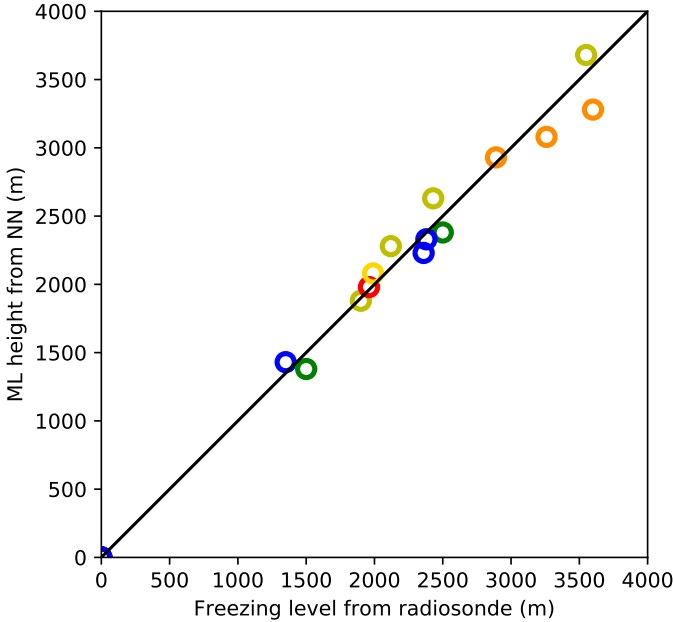

**Figure 8.** Comparison between ML height determined by NN and freezing level from radiosonde for Hohenpeißenberg. The colors denote
the time difference between the two heights: no difference (blue), 0–1 hour (green), 1–2 hours (olive green), 2–3 hours (yellow), 3–4 hours
(orange), 4–5 hours (red).

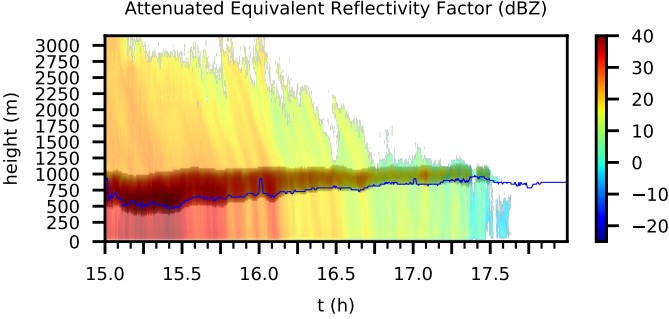

**Figure 9.** Comparison between ML height determined by NN and by a cloud radar, for Elmshorn on 22 Dec 2018, 15:00 to 18:00 UTC. The
colors in the background show the ZEA from the MRR, the gray shading denotes the area where the NN detected an ML, and the blue line
is the ML detected by a cloud radar.





Overall, the NN was well able to detect the correct ML height. Some weather situations are more difficult for the NN to handle than other situations. Cases where an ML is located well above the ground and fairly continuous in time and space pose the easiest cases and therefore the NN handles them best. The most frequent situations in which the NN is not able to capture the ML height well are those with discontinuous precipitation associated with convection. Then, also the human eye can hardly

detect the ML height in the measurement data. These situations are more frequent in summer due to stronger and more frequent convection in this season. Also, in some situations with snowfall, the NN detects clutter near the ground (wrongly detected MLs with a low probability and a small vertical and temporal extent), which is not covered by the suppression of clutter in the post-processing. More uncommon situations where the NN is sometimes not able to correctly detect an ML are situations when (i) there is strong horizontal wind, (ii) the signal directly below the ML is weak, (iii) there are disturbances in the signal,

or (iv) the dealiasing of the fall velocity failed. Horizontal wind leads to slanted structures of precipitation, which violates the assumption of the NN that an ML can be derived from characteristics within one vertical profile. Though aliasing of the fall velocity is corrected, this algorithm is not able to handle rare situations where the fall velocity is shifted by large amounts due to strong convection.

To ensure that the NN is able to handle different averaging intervals of the input data and different settings of the vertical

resolution of the MRR, sensitivity studies were made. These tests showed that the NN is well able to handle temporal averaging up to one minute and different vertical resolutions of up to 100 m. Also, a comparison to radiosonde data was made at Hohenpeißenberg where one MRR was located and radiosondes were launched. Though there were only few simultaneous ML observations, the agreement of ML height was very good. In addition, the detected ML is consistent with measurements from a cloud radar.

The NN presented here was designed to be used operationally, being able to handle different weather situations and settings of the MRR. Some decisions were made during the development concerning the design of the NN and the desired output. One such choice is the broad width of the detected ML, encompassing the whole process of melting. While tuning the NN, a balance was sought between too many false positive detections and too many false negative detections. For some applications, few false positives might be more beneficial, while for other applications, few false negatives might be more important. The

decisions made for this NN were aimed to be valid for a broad range of applications.

This study showed that a neural network approach is well suited to detect the pattern of the ML from radar data. Though we have used the MRR, this approach can in principle be used by other types of radars as well. In contrast to deterministic methods (e. g., Fabry and Zawadzki, 1995; White et al., 2002; Cha et al., 2009; Perry et al., 2017), the NN is able to detect uncommon melting layers such as melting layers on the ground or two melting layers at once, and does not need to be given thresholds or

the shape of a profile with a melting layer in advance. A disadvantage of this method is that an NN lacks all physical basis and is thus not suited to study the physics within the melting layer. The time resolution of our approach can in principle be as high as the time resolution of the measuring radar, giving the possibility of detecting fast changes in the melting layer height, which can be important at, e. g., airports. Post-processing can include time-averaging and filtering of short ML occurrences, if a false positive detection is deemed worse than a false negative detection.





Future work could include using data from other climatic regions. Also, the preparation of the data such as the dealiasing of the fall velocity, or the post processing of the NN could be improved. The chosen algorithms for data preparation and post processing are designed to be fast enough for real time use within an MRR, therefore, their complexity was limited. Extending these algorithms might improve the performance of the NN.

5    Fields of applications for the presented NN are for example airports where reliable ML data could help to detect the danger of icing. The local data can also be used as supporting information to improve the quality of spatial ML data as provided by weather radars or numerical models.

*Author contributions.*   MB and PM designed the neural network. MB was the main writer of the manuscript.

*Competing interests.*   The authors declare that they have no conflict of interest.

10   *Acknowledgements.*   The authors wish to thank Gerhard Peters for his support in developing the neural network. We gratefully acknowledge Eckhard Lanzinger, Michael Frech, and Jörg Steinert from the German Weather Service for providing the MRR data and the data from the melting layer algorithm of the weather radars.





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
