# Peer review of "Detecting the Melting Layer with a Micro Rain Radar Using a Neural Network Approach"

_Atmospheric Measurement Techniques, 2019_

## Referee Comment (RC1) · Anonymous Referee #1 · 20 Oct 2019

The authors have shown that a suitably trained neutral network *can* identify the melting layer in MicroRainRadar data. They have yet to show whether this is useful in the context of previous work. In my view, the paper is incomplete in its current form.

General Comments:

There is no comparison between the new neural network method and established methods to determine the identification of the melting layer height in vertically-pointing radar data. Both the neural network method and most of the previous empirical methods use radar reflectivity and Doppler velocity. The new method needs to *demonstrate* that is a superior in some useful ways to previous methods. There is some

discussion that the neural network method can identify two melting layers whereas the previous methods could not but that circumstance is rare and it is not clear if it worked well in the example shown in Figure 6 (see 6) below). Currently the main comparison is with training and validation data sets determined by eye and entered by hand with a mouse.

A comparison to a previous empirical method from the literature needs to be added. The method used by Perry et al. (2017) with MRR data is relatively simple to implement: From Perry et al. (2017) "In our analysis, the bottom of the melting layer is identified as the most negative gradient in Doppler velocity in the profile and the top of the melting layer as the most negative gradient in reflectivity. To determine the top of the melting layer, the average hourly melting layer thickness (top minus bottom) was combined with the 1-minute values of melting layer in that hour. We elected to discard any melting layer height values greater than one standard deviation of the hourly mean."

Specific comments

1) Page 1 Lines 11-12, Please correct. As melting occurs the snow particle is not "coated" with liquid water, rather is contains a mixture of liquid droplets and ice (see Knight 1979, JAS). The coating idea was a simplification used by some investigators to model melting (e.g. Yokoyama and Tanaka 1984).

2) Page 1 line 14-15 "particle density decreases due to acceleration and the smaller size of particles" this implies the density of water changes…please correct.

3) Page 2 Lines 21-22 and Page 17 line 5-6. Please correct. Vertically pointing radar data cannot determine occurrence of supercooled rain (i.e. freezing rain and icing) without an accompanying temperature profile. The radar can determine where the melting takes place. Once liquid, the drops have identical observed radar properties whether or not they are supercooled.

4) Lines 14 and 1. Please clarify the details regarding how the lower and upper boundary of the melting layer were determined by different variables. The necessity to use these other variables appears to contradict later statements in relation to description of method later that says that only ZEA and VEL are used.

5) Section 3.3 Please clarify the radar resolution volume size in the vertical of the C-band radar resolution volume over the MRR. For 90 km distance this is likely more than 1 km spatial resolution.

6) Page 10, lines 7-10, Figure 6 Please show the temperature profile from Greifswald at 12 UTC as part of this figure and explain why at 12 UTC there are no melting layers shown, just before 12.5 UTC the radar data shows only one melting layer and it is not until about just after 13.5 UTC that two melting layers one aloft and one at the surface are evident.

7) please convert x-axis times in Figures 1-7 and 9 to UTC hours and minutes.

---

## Referee Comment (RC2) · Anonymous Referee #2 · 2 Nov 2019

The paper describes a technique for automated detection of the melting level in the atmosphere with a 24-GHz radar. This technique is indeed of great importance, e.g., for meteorological research and air safety. The paper is nicely written and its contents be understood easily. The new detection technique seems to be powerful and superior to prior approaches. It should be published, but I would like to see some important additional information in the manuscript:

General issues: In general, neural networks are used to solve problems which are too complex to be understood in a straight forward way. Also, neural networks are often applied when a solution is necessary but information about the underlying process is

irrelevant and/or not necessary. But for application in science, the functionality of any method must be traceable. Concerning the current paper, the authors should make an additional effort to lay open the interiors of the trained neural network that was used for melting layer detection. This could be done by publishing computer code or by giving a graphical representation of their neural network in the paper.

Specific issues:

- Which criteria have been applied when defining the vertical extent of the melting layer by eye?

- Discussion of the limits of the approach (starting p.9 line 13ff) is a bit confusing. The authors discuss that limitation arise in convective systems in which droplets are transported upwards. Figure 5 does not show such extreme conditions at Hohenpeißenberg. A radiosonde launched close to Munich at 12UTC of 31-08-2017 shows a melting level at around 4000m a.s.l.:

http://weather.uwyo.edu/cgi-bin/sounding?region=europe&TYPE=TEXT%3ALIST&YEAR=2017&MONTH=08&FROM=311

The misdetections of melting events must hence be intrinsic to the method rather than being a result of a complex meteorological situation. Isn't it more probable that the algorithm reacts sensitive to vertical changes in terminal fall velocity at the edges of the skewed fall streaks of rain droplets?

---

## Author Comment (AC1) · 17 Dec 2019

To the general comments:

While we find a comparison to other methods interesting, we consider this to be outside the scope of this work. A comprehensive comparison to other methods would need to incorporate some measurement of the melting layer height from an independent source at a similar height and time resolution which we do not have available for our datasets. In addition to this, any comparison would also suffer from ambiguity of choice of other algorithm.

[Figure]

We have shown comparisons to other methods of determining the melting layer and show cases that traditional decision tree algorithms that incorporate some kind of consensus step will fail to analyze.

To the specific comments:

Numbers 1 and 2 will be corrected.

3. we do not claim that the MRR can identify supercooled rain. Page 2 Lines 21-22 does not refer to this. The text on Page 17 line 5-6 will be clarified.

4. The process of identifying the ML by eye will be explained in more detail. Using several variables while determining the ML by eye reduces the uncertainty involved in the process. In training the NN, it became clear that the quality does not increase by using more variables. This is plausible because some of the other variables are just derivatives of the reflectivity and fall velocity, which means they do not include any new information.

5. Yes, the vertical resolution of the C band radar over the MRR is around 700m for the closest radar and around 1500m for the farthest radar. However, since the model data from comso-d2 are also included in the algorithm and its vertical resolution is much higher, we do not feel that this information is important to mention. We added mentioning the model data in the description of the figure.

6. We added the temperature profile from Greifswald at 12 UTC. At 12 UTC, the temperature at the surface in Hamburg is about 1°C higher that in Greifswald (data from data climate center from DWD). The two locations are about 100km apart. Therefore, we do not expect the two melting layers to occur at the same time.

7. Axis labels will be changed.

---

## Author Comment (AC2) · 17 Dec 2019

We agree that the method should be traceable and will publish the complete parameter sets for the neural network combined with some sample python code showing preparation of the data and post processing.

To the specific issues:

1. The melting layer was determined by the following criteria:

- The extent of the maximum in reflectivity was determined using first and the second derivative of reflectivity - The acceleration of precipitation was used - Width of signal peak in spectrum shows strong gradients at the boundaries of the melting layer corresponding to quick changes in the attributes of precipitation - rain rate shows the strongest gradients at boundaries of the melting layer because the algorithm to retrieve rain rate in the micro rain radar misinterprets snow and melting particles very differently

We will extend our discussion of this in the manuscript.

2. It is indeed more probable that skewed rain streaks are responsible for not detecting the ML, though in some cases strong vertical winds distorting the ML might also be possible. We changed the explanation.

---

## Author Response (AR2)

**1. Referees' comments**

**a. Referee 1:**

A) My key comment from my original review was not addressed "The authors have shown that a suitably trained neutral network *can* identify the melting layer in MicroRainRadar data. They have yet to show whether this is useful in the context of previous work. In my view, the paper is incomplete in its current form. " I strongly disagree that such a comparison is outside the scope of work. The empirical algorithm used by Perry et al. (2017) is well less than 1 page of code and would be simple to implement and run on the data sets used in this paper. Without a comparison and proof that the neural network is superior method, this paper has no clear value. If someone proposed a new empirical method and did not show it was superior to previously published methods that paper would not have value either.

B) I do think it is important to mention the vertical resolutions of the C-band radars relative to the MRR since the C-band radars are being used to evaluate the algorithm in Section 3.3 and Figure 10. Please put in specific mention of the vertical resolution of the C band radar over the MRR is around 700m for the closest radar and around 1500m for the farthest radar. Please include this information in the paper.

C) Based on Figure 7, clearly the neural network algorithm does not work for short, isolated showers. I suggest that the output of the neural network be filtered to remove such short time interval data. Page 10 states that "During the training process, the NN was tuned to detect as little as possible in these cases, but it was not possible to train the NN not to detect anything without impeding the abilities of the NN for other cases. Therefore, convective cases are those cases where the NN has the most problems." There is nothing stopping the authors from filtering the output of the NN as part of the quality control. Including data that is clearly wrong substantially weakens the case for using this new algorithm.

D) I think there is still some miscommunication regarding the application to determining risk of icing. Aircraft icing requires high concentrations of supercooled cloud water. This can occur at heights above the 0 deg C level when air temperature is < 0 deg C OR when warm layers are present aloft where the ice melts and then closer to the ground the air temperatures are < 0 deg C. The detection of the melting layer informs about the melting of snow. The MRR data cannot detect cloud-size droplets nor address the temperature of the air. While above the melting layer, the air temperature < 0 deg C an be inferred, this is not the case for below the melting layer. Whether the melting layer represents a warm layer aloft with colder air at the surface or temperature decreasing with height is not known without a temperature profile. It is unclear how this method will help to assess the risk of icing at airports (page 2 line 2, page 18 line 25)

Minor comments
Page 7, line 28 "indizes" should be "indices"

**2. Responses**

**a. Referee 1:**

A. We have attempted to use the cited algorithm as a comparison, but it is not suitable for several reasons:
   a. The reviewer did not cite the complete algorithm used in the article. He left out several plausibility checks that turned out to be highly site specific and which will not

be applicable to a general algorithm. This is especially relevant since the algorithm was used on data recorded in the Peruvian Andes at a about 3500m above sea level.

b. The algorithm also fails to adequately react to situations in which there is precipitation but the melting layer is not present in the data. This is also probably caused by the specific measurement site and measurement parameters used in the article.

c. The final output shown in the article shows hourly median values of the melting layer height. The application of a median compared to a simple average hints at the presence of outliers in the time series provided by the algorithm. Our calculations also show quite a few outliers which substantiates this.

What is given in the paper seems to be a good way to analyze the data for this single measurement campaign, but not an algorithm suitable for general application. To get a fair comparison, we would need to develop the alternative algorithm to a state which at least replaces the site specific parts with a general applicable form and add quite a bit of plausibility checks to get around other problems, eg 2. above. This is not trivial work and cannot be done quickly.

Doing a comparison without this kind of preparation does not add any information to our own work.

B. We have added the information of the vertical resolution of the C Band radar for section 3.3. Figure 10 shows a comparison with a cloud radar, which has a much higher vertical resolution.

C. We agree that not filtering out data that is clearly wrong weakens the algorithm. However, we have experimented quite a bit with different ways of filtering out wrong results, during designing the neural network as well as using different filtering techniques on the output of the neural network. Since our algorithm is meant to be generally applicable, all cases need to be considered and concentrating only on filtering out bad results in case of showers results in less reliable results in other cases. We decided on a compromise which gives overall the best results for all situations. The method we presented in section 2.3 specifies the post processing and mentions how some wrongly detected cases of showers are filtered out.

D. We are aware that an MRR cannot replace the measurement of a temperature profile. We only suggest that the measurement of the ML can give more information, since measuring a melting layer results in the knowledge that falling frozen particles have melted at this height. The MRR cannot determine whether the molten particles refreeze or fall into layers with temperatures below 0°C. However, in combination with other measurements such as ground temperature the MRR can give valuable additional information, also if for example a melting layer is observed to decrease with height and might touch the ground in the near future and thus the end of an icing situation can be concluded. Especially if temperature profiles are not available, the MRR can add information to other measurements taken at airports to help judge the danger of icing.

**3. Changes**

P2L1   added clarification on MRR data only adding to the information necessary to judge danger of icing

P15L2 added information on vertical resolution of C band radars

P16L15 added information on vertical resolution of cloud radar

[revised manuscript text omitted]